# Interaction between Microsatellite Instability (MSI) and Tumor DNA Methylation in the Pathogenesis of Colorectal Carcinoma

**DOI:** 10.3390/cancers13194956

**Published:** 2021-10-01

**Authors:** Farzana Jasmine, Zahidul Haq, Mohammed Kamal, Maruf Raza, Gustavo da Silva, Katrina Gorospe, Rupash Paul, Patrick Strzempek, Habibul Ahsan, Muhammad G Kibriya

**Affiliations:** 1Institute for Population and Precision Health, Department of Public Health Sciences, Biological Sciences Division, The University of Chicago, Chicago, IL 60637, USA; fjasmine@health.bsd.uchicago.edu (F.J.); auck.gustavo@gmail.com (G.d.S.); kgorospe@uchicago.edu (K.G.); patrick.strzempek1@gmail.com (P.S.); hahsan@health.bsd.uchicago.edu (H.A.); 2Department of Surgery, Bangabandhu Sheikh Mujib Medical University, Dhaka 1000, Bangladesh; zhaq92@gmail.com; 3Department of Pathology, Bangabandhu Sheikh Mujib Medical University, Dhaka 1000, Bangladesh; kamalzsr@yahoo.com (M.K.); maruf_path@yahoo.com (M.R.); rupash_pal@yahoo.com (R.P.)

**Keywords:** MSI, colorectal cancer, interaction, CIMP, MMR, immune checkpoint inhibitor, *CTLA4*, *HAVCR2*

## Abstract

**Simple Summary:**

In colorectal cancer (CRC), mutations may occur in short, repeated DNA sequences, known as microsatellite instability (MSI). Tumor DNA methylation is another molecular change now recognized as an important biomarker in CRC. In a genome-wide scale, for the first time, we explored whether DNA methylation is associated with MSI status in CRC. We analyzed 250 paired samples (tumor and corresponding normal) from 125 CRC patients (m = 72, f = 53) at different stages. We found that many genes were methylated in tumor tissue compared to normal tissue. However, almost four times more genes showed such methylation changes in the tumor if the patient who also had MSI compared to patients without MSI. Our study shows an association of MSI and DNA methylation in CRC. The study also indicates an opportunity for potential use of certain immune checkpoint inhibitors (*CTLA4* and *HAVCR2* inhibitors) in CRC with MSI.

**Abstract:**

In colorectal cancer (CRC), the role of microsatellite instability (MSI) is well known. In a genome-wide scale, for the first time, we explored whether differential methylation is associated with MSI. We analyzed 250 paired samples from 125 CRC patients (m = 72, f = 53) at different stages. Of them, 101 had left-sided CRC, 30 had MSI, 34 had somatic mutation in KRAS proto-oncogene (*KRAS*), and 6 had B-Raf proto-oncogene (*BRAF*) exon 15p.V600E mutation. MSI was more frequent in right-sided tumors (54% vs. 17%, *p* = 0.003). Among the microsatellite stable (MSS) CRC, a paired comparison revealed 1641 differentially methylated loci (DML) covering 686 genes at FDR 0.001 with delta beta ≥ 20%. Similar analysis in MSI revealed 6209 DML covering 2316 genes. ANOVA model including interaction (Tumor*MSI) revealed 23,322 loci, where the delta beta was different among MSI and MSS patients. Our study shows an association between MSI and tumor DNA methylation in the pathogenesis of CRC. Given the interaction seen in this study, it may be worth considering the MSI status while looking for methylation markers in CRC. The study also indicates an opportunity for potential use of certain immune checkpoint inhibitors (*CTLA4* and *HAVCR2* inhibitors) in CRC with MSI.

## 1. Introduction

Chromosomal instability and microsatellite instability (MSI) are distinct, well-described pathways of colorectal carcinoma (CRC) [1,2]. MSI occurs in ~15% of colon cancers and is supposed to result from inactivation of the mutation mismatch repair (MMR) system by either MMR gene mutation or hypermethylation of the MMR genes, such as MutL homolog 1 (*MLH1*) promoter. MSI promotes tumorigenesis by generating mutations in target genes that possess coding microsatellite repeats. There are studies suggesting a link between biallelic methylation of the *MLH1* promoter and the development of MSI [3,4,5,6,7]. One study showed hypermethylation of the *p16* gene was found in 60% of MSI tumors compared to 22% in MSS tumors [8]. The same study also showed the association of hypermethylation of thrombospondin-1 (*TSP-1*), insulin like growth factor 1 (*IGF2*), and hypermethylated in cancer 1 (*HIC-1*) with MSI tumors [8]. In all these studies, the associations were tested in a handful of targeted genes. In general, in a non-metastatic setting, patients with MSI CRC have better prognosis [2,9]. However, in a metastatic setting, the presence of MSI may have poorer prognosis in CRC patients with metastasis, as has been seen in a recent meta-analysis [10].

Methylation of CpG islands is increasingly recognized as an important event in CRC [11,12,13,14,15,16,17]. The term CpG island methylator phenotype (CIMP) has been used to describe tumors in which some specific genomic regions are commonly methylated [18]. DNA methylation status can be considered as a useful predictor of post-surgical survival in CRC [19]. In the present genome-wide methylation study in humans, we explored whether the differential methylation of tumor DNA in CRC is associated with the MSI status of the tumor.

## 2. Materials and Methods

We carried out a genome-wide methylation assay (Illumina 450 K) for 250 paired samples from 125 CRC patients (m = 72, f = 53) at different stages (stage I: 25, stage II: 33, and stage III: 67). Of them, 101 had left-sided CRC (descending colon to rectum) and 30 had MSI, 34 had somatic mutation in *KRAS* (rs112445441), and only 6 had *BRAF* exon 15p.V600E mutation.

### 2.1. Tissue Samples

The fresh frozen samples were collected from 125 CRC patients (male = 72 and female = 53) at different stages (stage I: 25, stage II: 33, and stage III: 67) from the Department of Pathology, Bangabandhu Sheikh Mujib Medical University (BSMMU), Dhaka, Bangladesh, at different times, spanning between December 2009 and May 2016. During each collection period, all consecutive patients were selected. From each patient, the specimens were collected from the surgically resected tumor and the surrounding unaffected part of the colon about 5–10 cm away from the tumor mass. Surgical pathology fellow collected all samples from the operating room immediately after the surgical resection. Pathology was conducted independently by two pathologists and there was concordance in all 125 cases. Thus, from each individual, we obtained a pair of tumor and normal tissues, which were frozen immediately and shipped on dry ice to the molecular genomics lab, at the University of Chicago, for subsequent DNA extraction and methylation assay.

For each patient, we also abstracted key demographic and clinical data and tumor characteristics from hospital medical records. Written informed consent was obtained from all participants. The research protocol was approved by the “Ethical Review Committee, Bangabandhu Sheikh Mujib Medical University”, Dhaka, Bangladesh (BSMMU/2010/10096) and by the “Biological Sciences Division, University of Chicago Hospital Institutional Review Board”, Chicago, IL, USA (10-264-E).

### 2.2. DNA Extraction and Quality Control

DNA was extracted from fresh frozen tissue using the Puregene Core kit (Qiagen, Germantown, MD, USA). The average 260/280 ratio was 1.85. An electropherogram from the Agilent Bioanalyzer with Agilent DNA 12000 chip showed the fragment size to be >10,000 bp.

### 2.3. Genome-Wide Methylation Assay

We used 500 ng of 125 paired tumor and corresponding healthy tissue DNA for bisulfite conversion using an EZ-96 DNA Methylation Kit (Zymo Research, Irvine, CA, USA).

The HumanMethylation450 DNA analysis BeadChip v1.0 Assay kit was used (Illumina, San Diego, CA, USA). This chip presented 485,577 loci of which 150,254 in CpG Island, 112,067 in Shore (0–2 kb from island), 47,114 in shelf (2–4 kb from the island), and 176,112 in deep sea (>4 kb from CpG island). Paired samples (CRC and corresponding normal) were processed on the same chip to avoid the batch effect. From this assay, on average 17 loci per gene were interrogated. A Tecan Evo robot was used for automated sample processing and the chips were scanned on a single iScan reader. If the intensity of methylated loci is X and the intensity of unmethylated loci is Y, then the methylation score (beta value) is X/X + Y. If all are unmethylated (X = 0), then the methylation level is 0/0 + Y= 0. If all loci are methylated (Y = 0), then the beta value is X/X + 0= 1. If 50% probes are hybridized at methylated loci and 50% hybridized at unmethylated loci, then methylation score is 50/50 + 50 = 0.5.

### 2.4. MSI Detection

#### 2.4.1. Microsatellite instability (MSI) detection

A high-resolution melting (HRM) analysis method was used for detection of two mononucleotide MSI markers—BAT25 and BAT26 [13,20]. A tumor was defined as having MSI when it showed instability with at least one of these markers (BAT25 and BAT26), and as MSS when it showed no instability for both the markers. We confirmed the MSI using another novel marker CAT25 as well. We used published primer sequences [20]. Thermocycling and melting conditions were optimized for the CFX96 instrument and Bio-Rad Precision Melt Analysis software was used to identify MSI by differential melting curve characteristics [13]. Conventionally, the diagnosis of MSI in CRC is based on a set of five microsatellite markers (two mononucleotide and three dinucleotide repeats) proposed by the National Cancer Institute Research Workshop in Bethesda [21]. However, the original microsatellite panel has limitations resulting from the inclusion of dinucleotide markers, which are less sensitive and specific for detection of tumors with mismatch repair deficiencies. One of the suggestions was the exclusive use of mononucleotide repeats, improving the sensitivity of MSI detection in CRC [22]. Mononucleotide markers are more commonly quasi-monomorphic, potentially obviating the need to test the corresponding normal DNA [23]. BAT26 and BAT25, the best-known quasi-monomorphic mononucleotide repeats in the Bethesda panel, appear to undergo significant deletions in the large majority of tumors with MSI, proving to be very useful for the identification of MSI even without the use of corresponding germline DNA [24,25]. Among the other mononucleotide markers, Findeisen et al. [26] described a novel mononucleotide marker in the 3′ untranslated T25 region of the CASP2 gene (CAT25) that displayed a quasi-monomorphic repeat pattern in normal tissue and represented a highly promising candidate marker [26]. The efficiency of this CAT25 marker was also confirmed by other study [27]. 

The amplification conditions included the polymerase activation step at 95 °C for 2 min, followed by 5 cycles of denaturation at 95 °C for 15 s, annealing starting at 60 °C for 30 s, extension at 72 °C for 30 s, and an additional 33 cycles of denaturation at 95 °C for 15 s, annealing at 53 °C for 30 s, and extension at 72 °C for 30 s. Before the HRM step, the products were heated to 95 °C for 1 min and cooled to 40 °C for 1 min, to allow heteroduplex formation. HRM was carried out and the data collected over the range from 60 to 95 °C, with temperature increment of 0.2 °C/s at each 0.05 s. The BAT25 and BAT26 products were sequenced for validation. In this way, a total of 30 tumor samples showed MSI and all were confirmed by another relatively novel MSI marker CAT25 [26,27]. 

#### 2.4.2. KRAS and BRAF mutation detection

Tumor and adjacent healthy colonic tissue from 125 patients were tested for *KRAS* (rs112445441) and BRAF exon 15p.V600E mutation by high resolution melt analysis, as previously described [28].

### 2.5. Statistical Analysis

To compare the continuous variables (e.g., number of detected loci/samples or average signal intensity/average beta value, etc., between the two groups), we used one-way analysis of variance (ANOVA).

### 2.6. Genome-Wide Methylation Data Analysis

For measuring methylation, we used Illumina GenomeStudio software to generate the beta value for each locus from the intensity of methylated and unmethylated probes. The built-in control probes within the chip were used to normalize the intensity. The beta is calculated as:$$\frac{\mathrm{intensity}\;\mathrm{of}\;\mathrm{methylated}\;\mathrm{probe}}{\mathrm{intensity}\;\mathrm{of}\;\mathrm{methylated}\;\mathrm{probe}\;+\;\mathrm{intensity}\;\mathrm{of}\;\mathrm{unmethylated}\;\mathrm{probe}}$$

Hence, beta ranges between 0 (least methylated) and 1 (most methylated) and is proportional to the degree of methylated state of any particular loci. We exported the GenomeStudio generated beta-values to PARTEK Genomic Suite [29] for further statistical analyses.

The principal component analysis (PCA) and sample histograms were checked as a part of quality control analyses of the data. Mixed-model multi-way ANOVA (which allows more than one ANOVA factor to be entered in each model) was used to compare the individual CpG loci methylation data across different groups. In general, “tissue” (tumor/adjacent normal), MSI status (MSI/MSS), and tumor location (proximal colon/distal colon) were used as categorical variables with fixed effect since the levels “tumor/normal”, “MSI/MSS”, and “proximal/distal” represent all conditions of interest; whereas “person ID#” (as proxy of inter-person variation) was treated as a categorical variable with a random effect, since the person ID is only a random sample of all the levels of that factor. The method of moment estimation was used to obtain estimates of variance components for mixed models [30]. As per the study design, we processed both the CRC tissue and the corresponding adjacent normal sample from one individual in a single chip (one chip accommodates 8 samples). In the ANOVA model, the beta-value for the CpG loci was used as the response variable (Y), and “tumor” (tumor or normal), person ID#, “MSI-status”, and “location” were entered as ANOVA factors.

For paired analysis, we used the following model:$$Y_{ijk}=\mu +Tumor_{i}+Person_{j}+\varepsilon_{ijk}$$
where *Y_ijk_* represents the *k*-th observation on the *i*-th Tumor *j*-th Person. *μ* is the common effect for the whole experiment. *ε_ijk_* represents the random error present in the *k*-th observation on the *i*-th tumor *j*-th person. The errors *ε_ijk_* are assumed to be normally and independently distributed with the mean 0 and standard deviation *δ* for all measurements. Person is a random effect.

For the detection of interaction between the tumor and MSI, the following model was used:$$Y_{ijk}=\mu +Tumor_{i}+MSI_{J}+Tumor*MSI_{ij}+\varepsilon_{ijk}$$
where *Y_ijk_* represents the *k*-th observation on the *i*-th tumor *j*-th MSI. *μ* is the common effect for the whole experiment. *ε_ijk_* represents the random error present in the *k*-th observation on the *i*-th tumor *j*-th MSI. The errors *ε_ijk_* are assumed to be normally and independently distributed with mean 0 and standard deviation *δ* for all measurements.

To see if the interaction was present in both proximal and distal location, the following model was used:$$Y_{ijkl}=\mu +Tumor_{i}+MSI_{J}+Location_{k}+Tumor*MSI_{ij}+Tumor*Location_{k}+\varepsilon_{ijkl}$$
where *Y_ijkl_* represents the *l*-th observation on the *i*-th tumor *j*-th MSI *k*-th location. *μ* is the common effect for the whole experiment. *ε_ijkl_* represents the random error present in the *l*-th observation on the *i*-th tumor *j*-th MSI *k*-th location. The errors *ε_ijkl_* are assumed to be normally and independently distributed with mean 0 and standard deviation *δ* for all measurements.

In GO enrichment analysis, we tested if the genes found to be differentially methylated fell into a gene ontology category more often than expected by chance. We used the chi-square test to compare the “number of significant genes from a given category/total number of significant genes” vs. the “number of genes on chip in that category/total number of genes on the microarray chip”. Negative log of the *p*-value for this test was used as the enrichment score. Therefore, a GO group with a high enrichment score represents a lead functional group. The enrichment scores were analyzed in a hierarchical visualization and in tabular form.

## 3. Results

Patient characteristics are presented in Table 1. MSI was more frequently found in males than females (30% vs. 15%, *p* = 0.045) and on right-sided tumors compared to those on the left (54% vs. 17%, *p* = 0.0003).

### 3.1. Methylation Status of DNA Mismatch Repair (MMR) Genes in MSI and MSS Tumors

MSI is caused by impairment in DNA mismatch repair (MMR) genes. This impairment may be caused by mutation or promoter methylation. We examined if the CRC in our patients was associated with differential methylation of these genes. We examined 15 MMR genes—*MLH1*, MutL homolog 3 (*MLH3*), MutS homolog 2 (*MSH2*), MutS homolog 3 (*MSH3*), MutS homolog 6 (*MSH6*), O-6-methylguanine-DNA methyltransferase (*MGMT*), proliferating cell nuclear antigen (*PCNA*), PMS1 homolog 1 (*PMS1*), PMS1 homolog 2 (*PMS2*), DNA polymerase Eta (*POLH*), replication factor C subunit 1 (*RFC*), replication protein A1 (*RPA*), high mobility group box 1 (*HMGB1*), ligase-1 (*Lig1*), and microRNA 155 (*MIR-155*). There were a total of 370 loci for these 15 MMR genes in the methylation array used in this study. We had DNA samples from tumor tissue, and the adjacent (apparently healthy) colon tissue from 125 patients. First, we analyzed the paired (tumor–healthy) samples for MSI patients (*n* = 30) and MSS patients (*n* = 95) separately. The paired t-test results are presented in Figure 1A,B, respectively. Several loci (e.g., in *MGMT* and *MLH1* gene) had statistically significant *p*-values when the magnitude of difference was ignored. However, we could not find any locus in the MSI group or MSS group (out of the 370 loci within these 15 MMR genes), where the magnitude of differential methylation (delta beta) was 0.2 or more in either direction (hyper- or hypo-methylation) in CRC tumor tissue compared to corresponding healthy tissue. An example of *MGMT* is shown in Figure 2. For these 370 loci covering the MMR genes, person-to-person variation explained more than 62% of the variation in the methylation data, emphasizing the importance of paired analysis (corresponding tumor-healthy) to identify CRC-associated loci (see Figure 3).

In the next step, we also explored if the difference in methylation between the tumor and healthy tissue (delta beta) for any loci within the MMR genes was significantly different in the MSI tumors compared to the MSS tumors. There were several loci in the *MLH1* gene, which showed higher methylation in tumor tissue compared to healthy tissue in the presence of MSI, but again the magnitude of differential methylation was low. Therefore, our microarray-based methylation data did not have strong evidence to suggest that CRC was associated with marked differential methylation (at least 20%) of MMR genes.

### 3.2. Genome-Wide Differential Methylation in MSI Tumors

Paired comparisons of the 30 MSI CRC tissues to corresponding healthy colon tissues showed a total of 6209 differentially methylated loci (DML) significant at FDR 0.001 with a magnitude of difference of at least 20% (delta beta ≤ −0.02 or ≥0.02). This is shown in Figure 4B. These loci were associated with a total of 2316 genes.

### 3.3. Genome-Wide Differential Methylation in MSS Tumors

Similar paired comparisons of the 95 MSS CRC tissues to the corresponding healthy colon tissues showed a comparatively smaller number (*n* = 1641) of differentially methylated loci (DML) significant at FDR 0.001 with a magnitude of difference of at least 20% (delta beta ≤ −0.02 or ≥0.02). This is shown in Figure 4A. These loci were associated with 686 genes.

The results suggest that the presence of MSI in CRC may be associated with differential methylation (tumor vs. normal) in a large number of loci (covering larger number of genes). The Venn diagram in these two lists of DML in MSI tumors and MSS tumors (Figure 4C) suggest that 1388 DML were common in MSI and MSS tumors, while a large number of DML (*n* = 4821) were present in MSI and there were fewer DML (*n* = 253) present in MSS tumors.

### 3.4. Interaction of Tumor and MSI for Methylation

To focus on the interaction of the MSI and the tumor for differential methylation, we further examined the data using ANOVA model(s), where we included the interaction term “Tumor x MSI”. That way, we examined if the differential methylation (delta beta of tumor and normal) was different among the patients with or without MSI. The *p*-value of the interaction term indicates that there was 23,322 loci where the delta beta of the tumor and normal was statistically different among MSI and MSS patients at an FDR 0.05 level (shown in the right lower circle in the Venn diagram of Figure 5E).

The direction of the methylation change (up or down in the tumor) and the distribution of these 23,322 DML in the genome (relative to the CpG island) are shown in Table 2. The table shows that greater portions of the differentially methylated loci were near the CpG islands and shores.

The Venn diagram (Figure 5E) of these three lists is presented in the center of Figure 5. The intersections in the Venn diagram show that:

A. There were 1369 loci with no interactions, and these loci had at least 20% differential methylation in the tumor compared to normal, irrespective of whether the patient had MSI or not. Moreover, a representative locus from this group is shown in the upper left part (Figure 5A). It should be noted that the magnitude of difference was not statistically different among MSI and MSS patients.

B. There were 1301 loci with interactions and these loci had at least 20% differential methylation in the tumor compared to normal if the patient had MSI, and the magnitude of delta beta in MSI was significantly greater than the magnitude seen in MSS patients (see additional file 1: Appendix A). A representative locus from this group is shown in the upper right part (Figure 5B).

C. There were 19 loci with interactions and these loci had at least 20% differential methylation in the tumor compared to normal in both MSI and MSS patients (see additional file 2: Appendix A). However, the magnitude of delta beta was significantly different in the MSI and MSS groups. A representative locus from this group is shown in the lower right part (Figure 5D).

D. There were 46 loci with interactions and these loci had at least 20% differential methylation in the tumor compared to normal if the patient had MSS and the magnitude of delta beta in MSS was significantly greater than the magnitude seen in MSI patients (see additional file 3: Appendix A). A representative locus from this group is shown in the lower left part (Figure 5C).

### 3.5. Location of Tumor and Interaction of MSI with Tumor

In our study, similar to other studies, MSI was more frequently encountered in the right-sided tumors (see Table 1). We also noticed that differential methylation was more common in right-sided tumors. Therefore, in different ANOVA models, we also included the location of the tumor and additional interaction term “Tumor*Location” in the regression model. Even then, the interaction term “Tumor*MSI” remained significant at FDR 0.05 for a total of 9077 loci. In other words, many of the loci show higher differential methylation in the presence of MSI, irrespective of the location of the tumor.

### 3.6. Methylation of Some of the Previously Reported Genes

We also looked specifically at some other genes previously reported [8] to be associated with MSI (e.g., cyclin dependent kinase inhibitor 2A (*CDKN2A*, *p16*), thrombospondin-1 (*THBS1*)), but could not see marked differential methylation. However, our data support the previous finding of association of hypermethylation of the *HIC* gene and MSI in colon cancer (additional file 4: Appendix A).

We also looked at the commonly used genes for detection of CIMP phenotype (calcium voltage-gated channel subunit alpha1 G (*CACNA1G*), *CDKN2A*, cellular retinoic acid binding protein-1 (*CRABP1*), *IGF2*, *MLH1*, neurogenin-1 (*NEUROG1*), *RUNX* family transcription factor 3 (*RUNX3*), and suppressor of cytokine signaling-1 (*SOCS1*) [31]. In the 450 K chip used in this study, there were 107 loci covering four of these genes (*CRABP1*, *NEUROG1*, *RUNX3*, and *SOCS1*). When we looked at these 107 loci (Figure 6), a few, but not all of the loci in *SOCS1*, *RUNX3*, *CRABP1*, and *NEUROG1* showed a statistically significant interaction of MSI and the tumor, suggesting greater differential methylation in MSI tumors. Magnitude of differential methylation was not >20% in most cases. Examples are shown in Figure 7. It should be noted that we measured the methylation by probe-based detection of methylated bases in the microarray and it was clear that the degree of methylation was different in different genomic regions within the same gene; whereas most of the CIPM studies were conducted by PCR.

### 3.7. Possible Functional Prediction

Considering the distribution of the loci with interactions in relation to the genomic regions (enriched mainly in islands and shores), and the fact that close proximity of the methylated loci to the gene is more likely to affect the gene expression, for the functional prediction purpose, we restricted the list to include the differentially methylated loci in the CpG island associated to the promoter only (*n* = 60,100). There was a total of 264 loci (covering 138 genes) with strong interaction (MSI*Tumor Bonferroni *p* < 0.05). Of them, almost all (*n* = 262 DML, covering 137 genes) were hypermethylated in the tumor. The list of genes, covering these DML with strong interactions, shows enrichment of genes involved in fat digestion and absorption, autophagy, ABC transporters, PPAR signaling pathway, mTOR pathway, and prolactin signaling pathway (see additional file 8: Appendix A). The genes involved in the fat digestion and absorption were ATP-binding cassette transporter subfamily A member1 (*ABCA1*) and diacylglycerol O-acyltransferase 2 (*DGAT2*). The differential methylation and differential gene expression of these two genes are shown in additional file: Appendix A.

For functional prediction, we also used Reactome v76 (https://reactome.org/ Accessed 28 July 2021). In the Reactome Event Hierarchy, the list of hypermethylated genes presented enrichments for the immune system, DNA repair, and programmed cell death. The related pathways were, respectively, interleukin-7 (*IL7*) signaling; depurination and depyrimidination of damaged nucleotides; and activation of BH3-only proteins, which triggers apoptosis in response to developmental cues or stress-signals, such as DNA damage. Since genes related to these pathways were hypermethylated, in terms of gene expression, we assume that these pathways could be under-expressed, playing a role in the cancer progression. However, these are predictions only.

HumanBase (https://hb.flatironinstitute.org/ Accessed 28 July 2021) creates tissue-specific networks from data-driven predictions to describe gene function, regulation, expression, interactions, and diseases. Of the 138 unique, promoter associated genes, HumanBase indicated that the SRY-box transcription factor 4 (*SOX4*), ATPase plasma membrane Ca2+ transporting 4 (*ATP2B4*), CCAAT enhancer binding protein alpha (*CEBPA*), prostaglandin E receptor 4 (*PTGER4*), ATP binding cassette subfamily A member 1 (*ABCA1*) and polo-like kinase 3 (*PLK3*) genes play a role in colorectal cancer development. Our results showed that, out of those six genes, *PLK3* was the only hypomethylated gene. Hypomethylation of *PLK3* potentially indicates overexpression of this gene. Interestingly, HumanBase showed evidence of *PLK3* having a depletion effect on tumor suppressor phosphatase and tensin homolog (*PTEN*). Overexpression of *PLK3* may deplete *PTEN* levels, resulting in the inhibition of apoptosis and, ultimately, cancer progression.

### 3.8. Prediction of Gene Expression from Methylation Data

We examined if the DML showing statistically strong interaction in the presence of MSI (the 264 methylation loci covering 138 genes described above) also translated to gene expression. We had the gene expression data from a subset of these samples (first 75 tumors and 73 corresponding surrounding healthy colonic tissues). In the gene expression chip (Illumina HT12v4), we found a total of 187 probes covering 132 (out of 138) of the genes that showed differential methylation and strong interaction with MSI. Using a similar ANOVA model with an interaction term (Tumor*MSI), we found that statistically significant interactions (interaction *p* < 0.05) were found in 20 probes (out of 187) covering 17 genes (see additional file 5: Appendix A). In other words, predicting the biological function on a given pathway from the methylation data, based on the assumption that the hypermethylation would cause downregulation of the gene, may not be accurate. Moreover, we should note that the programs, such as Reactome and HumanBase, are (basically) based on gene expression data. Therefore, we admit that caution is needed for the interpretation of the predictions we have presented in the section above.

We also looked at the gene expression data of the MMR genes in the subset (as mentioned above). We could not see differential expression of *MLH1* in MSI or MSS tumor tissues compared to corresponding healthy tissues (see additional file 6: Appendix A and Additional File 7: Appendix A and Figure 8). GO-ANOVA suggested that, on average, the MMR genes were overexpressed 1.09-fold (CI 1.06–1.12) in MSI and 1.08-fold (95% CI 1.06–1.09) in MSS tumors compared to the corresponding healthy colon tissues.

### 3.9. Analysis of this Genomic Data in the Light of Immuno-Therapy

In 2017, the Food and Drug Administration approved the use of two immune checkpoint inhibitors (ICI), pembrolizumab and nivolumab, for MSI metastatic CRC. In that context, we tried to explore if the methylation and/or gene expression data and their interactions with MSI presented in this paper can be used to better understand the potential use of these ICIs in colon cancer. However, we acknowledge the fact that we did not have Stage 4 patients and we do not have clinical follow-up data. We looked at the five immuno-target genes—programmed cell death protein 1 (*PDCD1*) (also known as *PD-1 and CD279*), programmed death ligand 1 (*PDL1*) (also known as *CD274*), cytotoxic T-lymphocyte associated protein 4 (*CTLA4*), lymphocyte-activation gene *3* (*LAG3*), and T cell immunoglobulin and mucin-containing protein 3 *(TIM3*) (also known as hepatitis A virus cellular receptor 2 (*HAVCR2*)).

#### 3.9.1. Methylation data for ICI target genes

These immune target genes were represented by 49 probes in the microarray chip we used. The magnitude of hypermethylation of *LAG3* in CRC was higher in MSI compared to MSS. That also corresponded to the downregulation of the *LAG3* gene in CRC from the gene expression data (see Figure 9). Potential interpretation may be that the *LAG3* inhibitors are less likely to be effective in CRC. Multiple probes in *PDCD1* were hypomethylated in the tumor, more so in MSS.

#### 3.9.2. Gene Expression data for ICI target genes

We observed statistical significant interaction for differential expression of *HAVCR2* (interaction *p* = 0.017), *PDCD1* (interaction *p* = 0.026) and *CTLA4* (interaction *p* = 0.038) (see Figure 10). We found that *CTL4* was overexpressed by 1.399-fold (95% CI 1.044–1.873) in MSI tumors compared to corresponding normal colon tissues, whereas there was non-significant change in MSS tumors, perhaps indicating a potential beneficial effect of the *CTL4* inhibitor, such as ipilimumab, in MSI CRC only. In the same line, *HAVCR2* was overexpressed by 1.338-fold (95% CI 1.006–1.779) in MSI tumors compared to corresponding normal colon tissue, whereas there were non-significant changes in MSS tumors, perhaps indicating a potential beneficial effect of *HAVCR2* inhibitor in MSI CRC only (see Figure 9). For *PDCD1*, there was no differential expression in MSI, but it was slightly downregulated in MSS (fold change = −1.09 (95% CI −1.149 to −1.045)).

ICI acts through activating cytotoxic T lymphocytes (CTLs) that use the *Fas-FasL* pathway. Thus, we also looked at the *Fas* expression. *Fas* was downregulated (fold change = −1.283 (95% CI −1.593 to −1.086)) in MSI tumors compared to the corresponding normal colon tissues. It was slightly more downregulated (fold change = −1.59 (95% CI −1.737 to −1.458)) in MSS tumors compared to corresponding normal colon tissues (Figure 10).

## 4. Discussion

Previous studies addressing the methylation of a handful of genes indicated that MSI might be associated with methylation of some genes, such as *TSP1*, *IGF2*, *HIC-1*, etc. [5,8]. In the present study, on a genome wide scale, we report the interaction of MSI and tumor for differential methylation of tumor DNA in CRC. Our data suggest that, compared to the MSS tumor, in CRC, MSI tumors are associated with differential methylation of a much larger number of genes. Although some of the MMR genes were statistically more methylated in the tumor, the magnitude of differential methylation was very small. Therefore, we could not comment if hypermethylation of the MMR gene caused the MSI. We did not have somatic mutation data for the MMR genes to comment if the MSI was caused by mutation of the MMR genes. More importantly, the association does not mean causality. However, regardless of the initiating factor for the MSI, our data clearly shows that marked differential methylation (mostly ≥20% more methylation) is seen in many more genomic regions in MSI tumors compared to MSS tumors in CRC. *BRAF* mutation was present in only six cases and so the sample size did not allow us to analyze the samples with or without BRAF mutation. Moreover, in this series, the MSI was more frequent in males.

For the evaluation of CIMP, Tapial et al. examined the methylation status of the promoter regions of eight genes—*CACNA1G*, *CDKN2A*, *CRABP1*, *IGF2*, *MLH1*, *NEUROG1*, *RUNX3*, and *SOCS1*) [24]. Each patient was classified as CIMP-(+) or CIMP-(−), depending on whether tumors showed ≥5/8 or ≤ 5/8 [32].

Using 920 CRC tissues, Shuji Ogino et al. ranked the markers in the order of *RUNX3*, *CACNA1G*, *IGF2*, *MLH1*, *NEUROG1*, *CRABP1*, *SOCS1*, and *CDKN2A* [23]. After validating, they showed that a panel of markers, including at least *RUNX3*, *CACNA1G*, *IGF2*, and *MLH1* can serve as a sensitive and specific marker panel for CIMP-high cases [31].

Sun Lee et al. showed that CIMP-high CRC had a close association with high MSI (*p* = 0.031): 23.8% of CIMP-high CRC were MSI high and 52.6% of MSI-high CRC were CIMP high. High CIMP was associated with BRAF mutation (*p* = 0.012), whereas there was no association between CIMP and *KRAS* mutation [33].

However, there is a study showing the lack of association of methylation and MSI. Yu Luo tested 110 CRC samples in a Chinese population, where 11 cases (10%) were CIMP-H, 92 cases (83.64%) were CIMP-L, and 7 cases (6.36%) were CIMP-0. Moreover, 10 cases (9.09%) were MSI-H, and 100 cases (90.91%) were MSS and MSI-L [34]. The mutation rates of *KRAS*, NRAS proto-oncogene (*NRAS*), and *BRAF* genes were 50% (55 cases), 6.36% (7 cases), and 5.45% (6 cases), respectively. There was no significant association between the CIMP group and MSI group (*p* = 0.734). Moreover, no significant differences were found in the mutations between the three subtypes of the CIMP group and the *KRAS*, *NRAS* genes (*p* > 0.05), while there was a statistically significant difference among the three subtypes and the *BRAF* gene mutations (*p* < 0.0001).

Jeong Bae et al. studied MSI in 72 CRC patients and found 25% are CIMP+ who had had a later age of onset and poor differentiation along with pathological differences than CIMP-CRC [35].

Kawasaki et al. examined the relationship between insulin-like growth factor binding protein 3 (*IGFBP3*) methylation, *p53* expression, CIMP, and MSI in 902 population-based colorectal cancers [36]. *IGFBP3* methylation was far more frequent in non-MSI-high CIMP-high tumors (85% = 35/41) than in MSI-high CIMP-high (49% = 44/90, *p* < 0.0001), MSI-high non-CIMP-high (17% = 6/36, *p* < 0.0001), and non-MSI-high non-CIMP-high tumors (22% = 152/680, *p* < 0.0001). Among CIMP-high tumors, the inverse relationship between MSI and *IGFBP3* methylation persisted in p53-negative tumors (*p* < 0.0001).

In past, using a lower density array (27 k) in a smaller number of cases, we identified a large number of differentially methylated genes in CRC [13]. In terms of genomic regions that are methylated in CRC, our present study provides extensive research for the association of DNA methylation and MSI in CRC. To our knowledge, this is the first study to examine such interaction in CRC at a genome-wide scale, especially in a Southeast Asian population. Many of these loci would have been missed in combined analysis of MSI and MSS cases, especially if only a small number of MSI tumors are included in the mixed pool of CRC and the interaction is not considered. The present study only included a Southeast Asian population; we will have to examine if this can also be replicated in other populations.

Immune checkpoint inhibitor cancer immunotherapy has shown efficacy in various human hematological malignancies and solid tumors. The *PD-1* inhibitors—pembrolizumab and nivolumab—led to a durable response in some patients with previously treated MSI-H–dMMR metastatic colorectal cancer, a finding that contributed to Food and Drug Administration approvals of pembrolizumab and nivolumab for patients with MSI-H–dMMR metastatic colorectal cancer that has progressed after treatment, with fluoropyrimidine, oxaliplatin, and irinotecan [37,38,39,40]. A recent study shows promising results of pembrolizumab in MSI-high advanced CRC [41]. A study showed that some immunotherapeutic targets were found highly expressed in *BRAF* mutated patients [42]. We did not have any patient treated with ICI drugs; however, our molecular data suggested significantly increased expression of *CTLA4* and *HAVCR2* in only MSI tumors and not in the MSS tumors, suggesting possible beneficial effects of the *CTLA4* inhibitor (e.g., ipilimumab) and the *HAVCR2* inhibitor in this subgroup of CRC patients. Ipilimumab was granted accelerated approval for use in combination with nivolumab for advanced MSI-H or dMMR metastatic CRC. There are many *HAVCR2* inhibitors in the pipeline that are being tested in clinical trials, in different human cancers (BMS-986258 (Bristol-Myers Squibb, New York, NY, USA), TSR-022 (Tesaro, Waltham, MA, USA), LY3321367, and LY3415244 (Eli Lilly and Company, Indianapolis, IN, USA); INCAGN02390 (Incyte, Wilmington, DE, USA), MGB453 (Novartis, Basel, Switzerland), Sym023 (Symphogen A/S, Copenhagen, Denmark), RO7121661 (Hoffmann-La Roche, Basel, Switzerland), BGB-A425 (BeiGene, Peking, China)) are currently ongoing (ClinicalTrials.gov Identifiers: NCT03489343, NCT03680508, NCT02817633, NCT03099109, NCT02608268, NCT03652077, NCT03066648, NCT03446040, NCT03708328, NCT03311412, NCT03744468, NCT03752177, NCT0 3940352, NCT03307785). Initial results of these studies are expected in the future. However, similar to other ICI treatments, not all patients may respond to the *HAVCR2* blockade. Our findings of overexpression of *HAVCR2* and *CTLA4* in MSI CRC may justify the molecular basis of such ICI treatments in this subgroup of CRC patients provided there is clinical indication. A recent study points toward an epigenetic regulation of HAVCR2 [43].

A study reported that colon cancer cells with lower *Fas* expression levels exhibit decreased sensitivity to FasL-induced apoptosis [44]. Our data also show lower expression of *Fas* in MSS, which is in accordance with the findings of other trials, showing a lack of beneficial effects of ICI in MSS colon cancer.

## 5. Conclusions

Our genome-wide methylation study shows, for the first time, evidence of association between MSI and tumor DNA methylation in the pathogenesis of CRC. Given the interaction seen in this study, it may be worth considering the MSI status while looking for differential methylation markers in CRC. The study also showed an increased expression of *CTLA4* and *HAVCR2* in CRC in the presence of MSI, suggesting an opportunity for potential use of certain checkpoint inhibitors (*CTLA4* and *HAVCR2* inhibitors) in CRC with MSI.

## Figures and Tables

**Figure 1 cancers-13-04956-f001:**
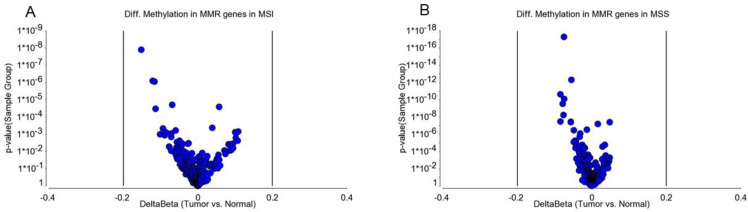
Differential methylation of MMR genes in MSI and MSS samples compared to the corresponding healthy colon tissue. The volcano plot shows the delta beta of 370 loci covering 15 mismatch repair genes (MMR) on the x-axis and the *p*-value on the y-axis for (**A**) MSI and (**B**) MSS tumors.

**Figure 2 cancers-13-04956-f002:**
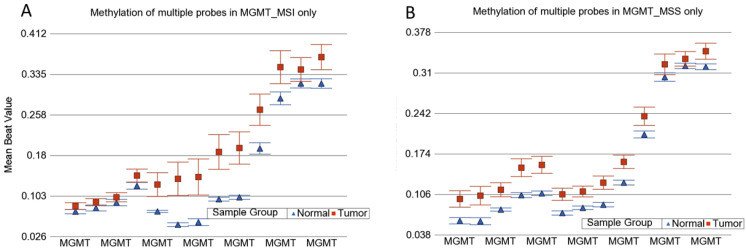
*MGMT* methylation status in MSI and MSS samples. Methylation status of multiple loci in the *MGMT* gene in MSI and MSS CRC (in red) compared to the corresponding normal colon tissue (in blue). Changes in MSI are shown in (**A**), and changes in MSS ate shown in (**B**).

**Figure 3 cancers-13-04956-f003:**
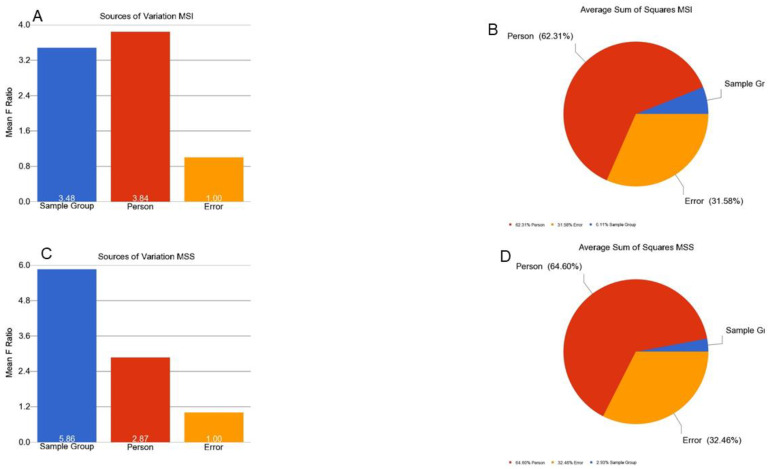
Variations of beta values in MMR genes in MSI and MSS samples. The source of variation in the beta value that can be explained by factors in the ANOVA models are shown in the figure. The mean F-Ratio (F-statistics for the factor/the F-statistics for the model error) representing the significance of the factor in the ANOVA model are shown in the bar graph (**A**,**C**). The sum of squares in the ANOVA model representing the proportion of the variation explained by the factors are shown as the pie chart (**B**,**D**). The proportion of variation that can be explained by the “person-to-person variation” is shown in red; the proportion that can be explained by the “sample group (tumor or normal)” is shown in blue; the proportion that could not be explained by the ANOVA model (the “error”) is shown in orange. The variation in MSI is shown in the top panel (**A**,**B**) while the variation in MSS is shown in the bottom panel (**C**,**D**).

**Figure 4 cancers-13-04956-f004:**
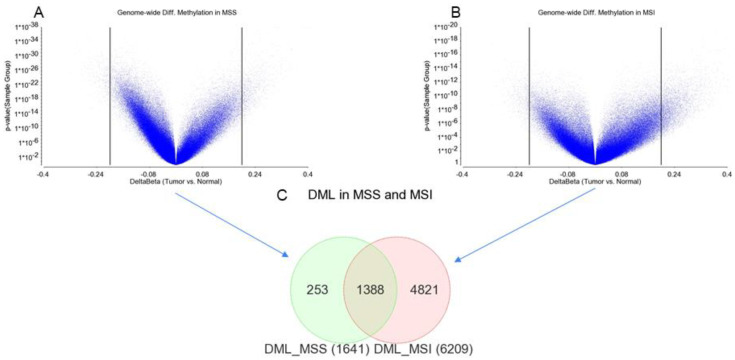
Genome-wide DML in MSS and MSI tumor tissue compared to corresponding healthy colon tissue. Differential methylation in MSS and MSI are shown in (**A**,**B**), respectively. The differential methylation between the tumor and normal (delta beta) is shown on the *x*-axis and the *p*-value for the paired comparison between the tumor and normal tissue is shown on the *y*-axis. (**C**) The overlap between the differentially methylated loci in MSS (*n* = 1641) and MSI tumors (*n* = 6209) are shown in the Venn diagram.

**Figure 5 cancers-13-04956-f005:**
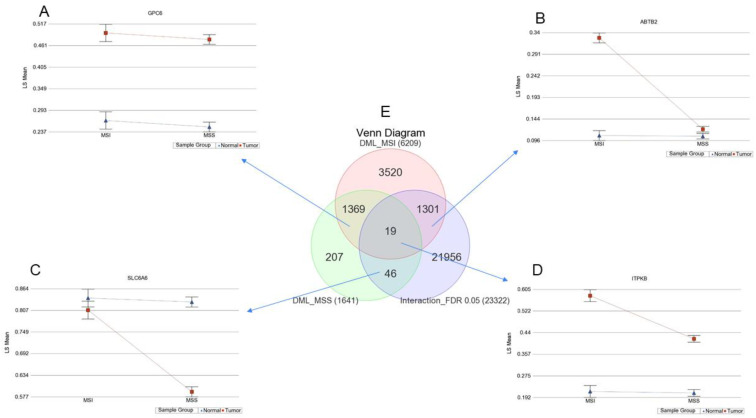
DML among MSI and MSS patients and different forms of interaction: (**A**) shows an example of the glypican 6 (*GPC6*) gene with no interactions where the tumor tissue shows significantly higher methylation compared to normal in both MSI and MSS tumors, and the magnitude of the difference is almost equal. (**B**) Shows an example of ankyrin repeat and BTB domain containing 2 (*ABTB2*) gene where the tumor tissue shows marked hypermethylation in MSI tumors, but not that much in MSS. (**C**) Shows an example of the inositol-triphosphate 3-kinase B (*ITPKB*) gene, which is hypermethylated in tumor tissue irrespective of MSI status, but the magnitude of the difference is higher if the tumor is MSI compared to when the tumor is MSS. (**D**) Shows an example of the solute carrier family 6 member 6 (*SLC6A6*) gene, which is markedly hypomethylated in MSS tumors and the magnitude is significantly more in MSS than MSI. (**E**) Venn diagram showing the overlap between the lists of DML in MSI, MSS, and the list of loci showing significant interaction between tumor and MSI status (interaction *p*-value FDR 0.05).

**Figure 6 cancers-13-04956-f006:**
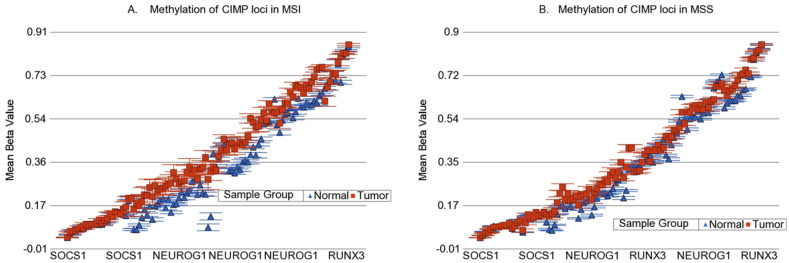
Methylation status of CIMP genes in the tumor (red) and normal (blue) in MSI and MSS. All 107 loci covering these genes commonly used for the detection of CIMP phenotype are shown. Changes in the MSI and MSS are shown in (**A**) and (**B**) respectively.

**Figure 7 cancers-13-04956-f007:**
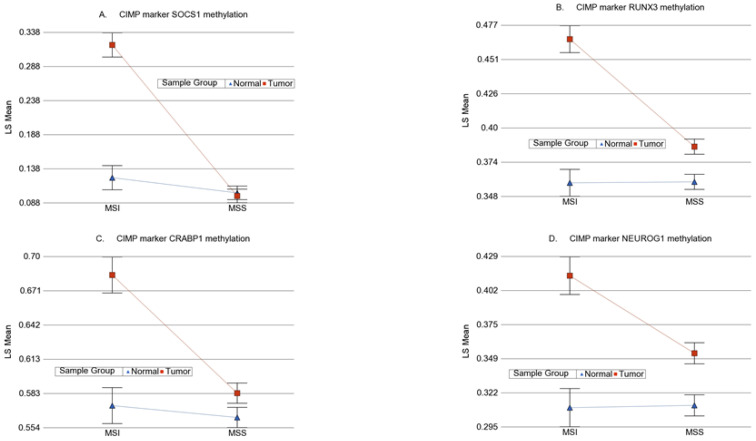
Examples of the interaction of MSI with differential methylation in some of the reported CIMP genes. Each figure suggests a greater differential methylation in MSI tumors (shown in red) compared to corresponding healthy tissue (shown in blue), but in most cases, the magnitude of differential methylation did not exceed 20%. The methylation changes in *SOCS1*, *RUNX3*, *CRABP1* and *NEUROG1* genes are shown in (**A**), (**B**), (**C**) and (**D**) respectively.

**Figure 8 cancers-13-04956-f008:**
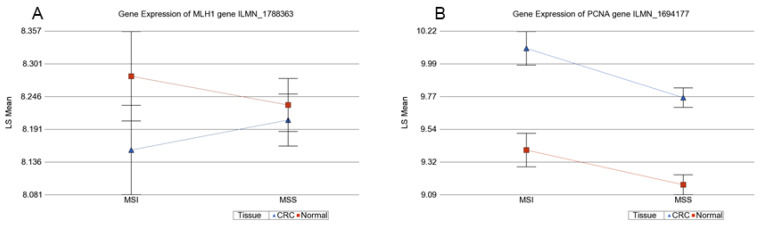
Differential gene expression of *MLH1* and *PNCA* in the tumor (in blue) compared to normal tissue (in red). (**A**) Shows no differential expression of the *MLH1* gene in MSI and MSS. (**B**) Shows an overexpression of *PCNA* in the tumor, in both MSI and MSS.

**Figure 9 cancers-13-04956-f009:**
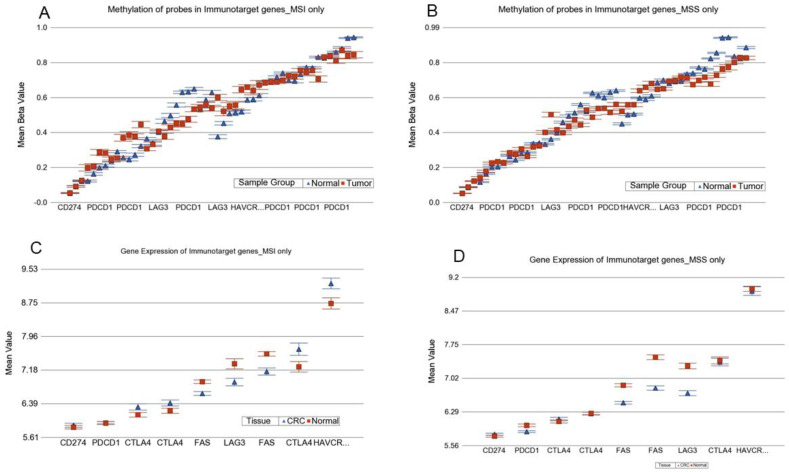
Differential methylation and gene expression of immune target genes in MSI and MSS colon cancer. In the upper panel, Figure (**A**,**B**) show multiple probes for differential methylation on the x-axis and the beta value on the y-axis for MSI and MSS CRC, respectively. In the lower panel, Figure (**C**,**D**) show multiple probes for gene expression on the x-axis and the log_2_-transformed expression value on the y-axis for MSI and MSS CRC, respectively.

**Figure 10 cancers-13-04956-f010:**
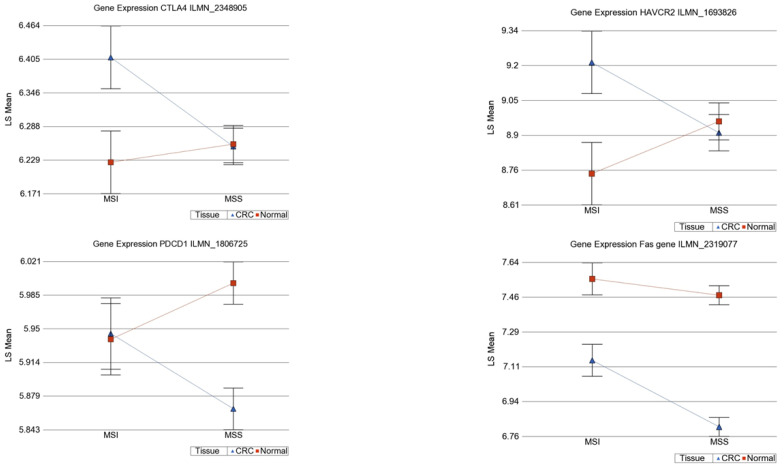
Interaction of MSI status and gene expression pattern of immune target genes.

**Table 1 cancers-13-04956-t001:** Patient characteristics. Bold was used to show the statistical significance.

Characteristics	Category	MSS	MSI	*p*-Value
Sex	Male	50	22	**0.045**
	Female	45	8	
Age	mean	46.08	45.63	0.879
	(SD)	(14.86)	(11.64)	
*KRAS*	Wild	70	21	0.689
	Mutant	25	9	
*BRAF* ^V600E^	Wild	90	28	
	Mutant	4	2	**0.048**
Location	Left	84	17	**0.0003**
	Right	11	13	
Pathology	Adenocarcinoma	81	13	0.806
	Mucinous adenocarcinoma	26	4	
Stage	Stage-1	17	8	0.227
	Stage-2	23	10	
	Stage-3	55	12	

**Table 2 cancers-13-04956-t002:** Distribution of the 23,322 DML showing interactions with MSI by relation to CpG island.

Relation of the Loci to CpG	Hypomethylated in Tumor	Hypermethylated in Tumor	# of DML	% of DML	# of Loci in Chip	% of Loci in Chip
Island	533	7837	8370	35.89	150,254	30.94
North Shelf	309	309	618	2.65	24,844	5.12
North Shore	1245	3873	5118	21.94	62,870	12.95
South Shelf	261	258	519	2.23	22,300	4.59
South Shore	939	3137	4076	17.48	49,197	10.13
Deep Sea	1975	2646	4621	19.81	176,112	36.27
Total	5262	18,060	23,322		485,577	

## Data Availability

All supporting data are presented in the tables presented in the main manuscript and as additional material.

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
