# Peer review of "Interaction between Microsatellite Instability (MSI) and Tumor DNA Methylation in the Pathogenesis of Colorectal Carcinoma"

_cancers, 2021, doi:10.3390/cancers13194956_

Round 1
Reviewer 1 Report
The manuscript entitled "Interaction of Microsatellite Instability (MSI) and tumor for Differential DNA Methylation in Colorectal carcinoma" highlighted an association between MSI and tumor DNA methylation in the pathogenesis of CRC.
- The Authors should better explain why they adopted only BAT25 and BAT26 panel for MSI evaluation and no the Bethesda panel or the five mononucleotic panel.
- Gene acronyms should be written in italics.
- The Authors should provide the expand forms for all acronyms, including gene acronyms, through the text when they first appear.
- Mutations should be reported as follow: e.g. BRAF exon 15 p.V600E.
Reviewer 3 Report
The authors performed genome-wide differential DNA methylation study, to explore whether the methylation of tumor DNA is associated with the MSI status of colorectal cancer (CRC). They have analyzed tumor tissue and normal sample from 125 CRC patients in Bangladesh, and consequently found the evidence of association between MSI and tumor DNA hyper-methylation. Their detailed analysis focusing MMR genes did not suggest that CRC was associated with differential methylation of the genes.
The result confirmed the previously reported evidence that CRC with MSI-high, CIMP high, hyper mutation are classified in consensus molecular subtype (CMS)1.
This study is well examined but there are still some queries to answer.
- A reviewer found it quite interesting that the differentially methylated loci (DML) with strong interaction with CRC shows enrichment of genes involved in fat digestion and absorption (Figure S1). The enrichment analysis comparing MSI and MSS should be quite impressive for readers.
- Although the discussion part is well written, the conclusions are quite common. Please emphasize the novelties of this study in conclusion and abstract.
Reviewer 4 Report
Jasmine et al. provide a study on correlation of the genome-wide methylation status and MSI in colorectal cancer (n=125). The authors suggest taking MSI status into account while looking for differential methylation markers in CRC.
The manuscript is very well-written. Nevertheless, we would like to present our critics in order to further improve the scientific quality of the work.
Major comments:
- The title is somewhat misleading. We would suggest rephrasing it. For example, ‘Interaction between Microsatellite Instability and tumor DNA methylation regulates the pathogenesis of colorectal carcinoma’.
- Figure 3: the pie charts may not be an ideal way to present the obtained statistical data. Please think about a more appropriate way to illustrate your data. Please kindly add a short description for “person”, “error” and “sample group” since this will markedly help the reader to quickly capture the figure’s main statement.
- Line 119: Wrong reference [13]. Please kindly check and provide correct citation.
- “Results” section is too long in relation to other sections (especially the Discussion)
- Please add survival / prognostic data if available.
Minor comments:
- The abstract is precise and well-structured. You may remove the headings (e.g., “Background”, “Methods”) from the abstract itself.
- Line 42: “Some study showed…” better change to “One study showed…”
- The introduction is well-written. Nevertheless, we suggest dividing the introduction in two paragraphs (MSI/ Methylation) in order to improve the structure of this section
- Line 53: We would be keen on to know more about the association of DNA methylation status and post-surgical survival time.
- Line 66: Please kindly define “collection period” and “all consecutive samples”. Does the latter mean tumour and adjacent healthy tissue?
- Line 71: “histopathologists” = “pathologists”?
- Line 120 (Statistical analysis): Please kindly provide the software you have used, define significant p-value(s).
- Line 182 (Results): You may also provide p-values here.
- Line 199, 208: MGMT – italics for human genes.
- Fonts used in the figures are too small and not well readable. Please adjust them. In addition, for Figure 1, 2, 3, 4, 5, 6, 7, please kindly provide higher resolution images.
- Line 236: Instead of “upper left volcano plot” use “Fig. A”
- Please shorten the method part on Genome-wide Methylation data analysis or place in it the supplement section.
Reviewer 5 Report
The authors present an interesting study on the role of microsatellite instability and its putative interactions with DNA methylation.
First, the term interaction is misleading, as it implies a mechanism that is shown. The interaction in terms of the manuscript here is rather an association/co-detection which is empirically and statistcally correlated. This is not a negative critcisms as such research is also important, but needs to be clarified.
Second, the methods, especially the statistical approach needs to be explained in sufficient details or be cross checked by a mathematician or an expert in biometrics.
Third, some figures are not clear. As an example, figure 2 is nearby unreadable due to the font size, and one axis contains one parameter in several columns. This is not clear. Also the axes of figure 1 are too small to be readable, and this is true for all figures.
Fourth, it is not clear how the authors define MSS and MSI. In general colorectal carcinoma are grouped into these catergories by using the Bethesda (plus) panel, which embraces five plus five microsatellites. The authors here one the one hand talk about 15 microsatellites, and than refer to about 23000 sites for the whole analyses. This is poorly explained.
Reviewer 6 Report
This is an interesting study that aimed to investigate the association between DNA methylation is and MSI status in CRC, in genome-wide scale. In general, the manuscript is well written and the research field is promising. English language is fine; and check throughout the text for consistent use of abbreviations. Please, if appropriate, add a legend for table 2. I would suggest to include more recent references (published in the last three years).
Round 2
Reviewer 1 Report
I have no further comments.
Author Response
Thank you very much.
Reviewer 2 Report
The authors have answered the questions sufficiently. I have no further comments.
Author Response
Thank you very much. All the points were very helpful.
Reviewer 4 Report
We warmly thank Jasmine et al. for providing clarifications on our queries. Furthermore, we acknowledge the work as a great example of collaborative effort between Bangladesh, a country with limited scientific resource, and the USA offering to compensate that limitation. We wish the authors good luck in their future joint endeavors.
Please find below our feedback on the re-submitted manuscript:
- We thank the authors for correcting the reference number 13
- About the survival data that we asked from the authors: we accept the unavailability of the data. However, we would recommend keeping such record if possible for follow-up studies as it reflects a direct relation with disease progression and morbidity.
- The restructuring that we suggested for the introduction section (separate paragraphs for MSI/ Methylation) is well-implemented by the authors. However, it is not clear from the introduction whether the authors worked on animal models or human cohorts. We would recommend adding few relevant lines on study subjects.
- Regarding the terms “histopathologists” and “pathologists”, we suggest to uniformly use the term ‘pathologist’ throughout the manuscript to avoid any discrepancy or confusion.
- For figures and labeling, we would like to ask the authors to adjust them into bigger font sizes so that they read better. Following the journal's submission guideline might help in this instance.
- We respect the other reviewer’s view on the Genome-wide Methylation data analysis and also gladly accept the authors’ defense.
Reviewer 5 Report
My previous comments have been thouroghly addressed, thank you.
Author Response
Thank you very much